# Incidental Findings in Pediatric Patients: How to Manage Liver Incidentaloma in Pediatric Patients

**DOI:** 10.3390/cancers15082360

**Published:** 2023-04-18

**Authors:** Andrius Cekuolis, Dagmar Schreiber-Dietrich, Rasa Augustinienė, Heike Taut, Judy Squires, Edda L. Chaves, Yi Dong, Christoph F. Dietrich

**Affiliations:** 1Ultrasound Section, Department of Pediatric Radiology, Radiology and Nuclear Medicine Centre, Vilnius University Hospital Santaros Klinikos, 08661 Vilnius, Lithuania; 2Localinomed and Pediatric Department, Kliniken Salem, 3013 Bern, Switzerland; 3Children’s Hospital, Universitätsklinikum Dresden, Technische Universität Dresden, 01062 Dresden, Germany; 4Department of Radiology, UPMC Children’s Hospital of Pittsburgh, Pittsburgh, PA 15224, USA; 5Radiology Department, Hospital Regional Nicolas Solano, La Chorrera 1007, Panama; 6Department of Ultrasound, Xinhua Hospital affiliated to Shanghai Jiaotong University School of Medicine, Shanghai 200092, China; 7Department Allgemeine Innere Medizin (DAIM), Kliniken Hirslanden Beau Site, Salem und Permancence, 3013 Bern, Switzerland

**Keywords:** guideline, detection, characterization, hepatocellular adenoma, hepatocellular carcinoma, focal nodular hyperplasia, hepatoblastoma

## Abstract

**Simple Summary:**

The World Federation for Ultrasound in Medicine and Biology (WFUMB) has recently launched a project on incidental findings (IF). Incidental findings of almost all organs are common and the topic is, therefore, of importance. This is a new concept of papers following a strictly defined structure and including the following headings: prevalence, epidemiology, clinical significance, clinical scenarios and the role of ultrasound and other imaging methods, summary statements and management strategies. The organs so far discussed in published papers are the adrenal glands, scrotum, salivary glands, spleen, female pelvis, biliary tree and, additionally, the important topic of managing incidental findings reported by medical students performing educational ultrasound examinations. The current paper summarizes IF of the liver in pediatric patients with special attention to the use of radiation-free ultrasound and contrast-enhanced ultrasound (CEUS).

**Abstract:**

The World Federation for Ultrasound in Medicine and Biology (WFUMB) is addressing the issue of incidental findings (IFs) with a series of publications entitled “Incidental imaging findings—the role of ultrasound”. IFs in the liver of newborns and children are rare and much less commonly encountered than in adults; as a result, they are relatively much more frequently malignant and life-threatening, even when they are of benign histology. Conventional B-mode ultrasound is the well-established first line imaging modality for the assessment of liver pathology in pediatric patients. US technological advances, resulting in image quality improvement, contrast-enhanced ultrasound (CEUS), liver elastography and quantification tools for steatosis have expanded the use of ultrasound technology in daily practice. The following overview is intended to illustrate incidentally detected liver pathology covering all pediatric ages. It aims to aid the examiner in establishing the final diagnosis. Management of incidentally detected focal liver lesions (FLL) needs to take into account the diagnostic accuracy of each imaging modality, the patient’s safety issues (including ionizing radiation and nephrotoxic contrast agents), the delay in diagnosis, the psychological burden on the patient and the cost for the healthcare system. Moreover, this paper should help the pediatric clinician and ultrasound practitioner to decide which pathologies need no further investigation, which ones require interval imaging and which cases require further and immediate diagnostic procedures.

## 1. Introduction and Definition

The World Federation for Ultrasound in Medicine and Biology (WFUMB) is addressing the issue of incidental findings (IFs) with a series of publications titled “Incidental imaging findings—the role of ultrasound” [1,2,3,4,5,6,7,8]. Each WFUMB position paper on IFs will follow the same template and, accordingly, will be uniformly structured to help readers interpret the key messages [7], will discuss the prevalence and imaging features of hepatic incidental finding in pediatric patients and will discuss strategies for work- and follow-up. In addition, imaging examples will be provided.

## 2. Definition

An incidental finding (IF) during an imaging examination is the unintended and unexpected discovery of an abnormality that may be clinically significant. IFs on imaging studies are asymptomatic and unexpected pathologies, unrelated to the presenting illness. They often necessitate further imaging or diagnostic work-ups which entail both a psychological and an economic burden for the patient and the healthcare system [2,3,4,5,6,7,8,9,10].

### 2.1. Prevalence, Epidemiology

Liver incidentalomas are commonly encountered with a reported frequency of up to 33% of radiological studies and 50% of autopsies in adult patients [11,12], but they are rarely reported in children. The reasons are manifold and have been discussed in detail elsewhere [13,14,15,16,17,18].

### 2.2. Clinical Scenarios and the Role of Ultrasound and CEUS

Ultrasound is the primary modality for the evaluation of liver pathology in children, including liver incidentalomas (IF). Contrast-enhanced ultrasound (CEUS) was introduced more than twenty years ago. The most important indications for using CEUS include: the characterization of indeterminate focal liver lesions using conventional B-mode ultrasound, computed tomography and in staging, and follow-up of previously known focal liver lesions (FLL). The latter includes pediatric patients with known cancer, patients who are under a surveillance program for chronic liver disease or other conditions that predispose them to malignancy [16,17,19], and patients who have undergone interventional procedures (e.g., post chemotherapy, or ablation) [20,21,22,23,24].

The European approval study for SonoVue 2001 was published as a first prospective multicenter trial with convincing results [25]. In the following years, the use of SonoVue/Lumason has been widely studied and shown to be of value, as has been represented in prospective studies and meta analyses [26,27,28,29,30,31,31,32,33,34,35,36], in retrospective studies including some pediatric patients as well [37,38,39,40,41,42,43,44], and in evidence-based guidelines [15].

Given its well-established role in liver imaging and its advantages over CT and MR imaging—including better patient tolerability, better performance in the presence of parents or caregivers at the point of care (bedside), better performance at the time of primary detection with complete assessment and definitive diagnosis, reduced cost and lack of ionizing radiation, and lack of renal, hepatic, cardiac, cerebral and thyroid toxicity—it has an important role in the evaluation of FLL [15,45,46,47,48,49]. CEUS does not require sedation.

More than ten years ago, this project on incidental findings was initiated to, (1) carefully assess the pretest probability which included: presentation of the patient, medical history (including patient’s background risk of malignant and inflammatory liver disease), assess laboratory data and liver stiffness using elastography and (2) to assess lesion characteristics including size and echogenicity [50,51]. In addition, the psychological, healthcare and economic impact of every imaging modality should be taken into account for appropriate management, diagnosis and follow-up of these patients. The initial attempt already highlights an important concept for the management of every incidentally found FLL by including all clinical information available and by using a multiparametric imaging approach [50,51,52]. Patients should be subcategorized into no, low and high risk for significant pathology based on the clinical and imaging features.

Later, a similar approach was used for guidelines issued by the American College of Radiology (ACR) on the management of liver incidentaloma using CT [53]. It is of importance that the partial volume artifact limits the ability of CT to characterize lesions smaller than 1 cm [14,48,49,54]. This limitation is less applicable in US because modern US devices can adequately visualize even small lesions and characterize their echogenicity (down to 1 mm).

Patients with no risk include healthy subjects being examined for screening purposes, patients with low risk of malignancy and those with no history of primary malignancy, no hepatic dysfunction and no risk factors. Patients with a high risk are those with known malignancy that is associated with liver metastases, liver cirrhosis and other hepatic risk factors including hepatitis, non-alcoholic steatohepatitis, alcohol overuse, cholangitis, choledochal cysts, haemochromatosis and other hereditary hepatic conditions or anabolic steroid use [53]. According to the EFSUMB, WFUMB and the National Institute for Health and Care Excellence (NICE) guidelines, CEUS is recommended for the characterization of incidentally detected focal liver lesions in adult patients, if the unenhanced US is inconclusive [15].

The unique role of computed tomography for staging and magnetic resonance imaging for characterization and staging in oncological patients is undoubtedly accepted.

How to perform contrast-enhanced ultrasound—including patient preparation [46], ultrasound contrast agent dose [16,17,19], interpretation of imaging findings [45], ultrasound contrast agent pharmacodynamics [45,55,56,57], CEUS phases [55], artifacts [58,59,60], and terminology [45]—have been reported elsewhere and are not part of this paper. As for any finding during any ultrasound examination, an IF should be documented in two different imaging planes, if possible, with cine loops [7,8,61].

## 3. Situs Inversus, Size and Shape Variations

Complete or incomplete situs inversus are not frequent incidental findings during ultrasound. Situs inversus is a congenital abnormality in which the heart may be on the right side (“right-hearted”) and/or visceral organs are reversed or mirrored from their normal positions. The reported frequency is about 1/10.000. Most pediatric patients have no medical symptoms or complications; however, a higher rate of cardiac anomalies has been reported. In a few patients with situs ambiguous or heterotaxy, situs cannot be determined. In these patients, the liver may be midline, the spleen absent or multiple, and the bowel malrotated [62,63].

### 3.1. Hepatomegaly and Variants of Shape and Echogenicity

The measurement of liver size is not generally recommended due to large variations and non-reproducible techniques. The US practitioner must, however, use their judgement, incorporating patient factors, when interpreting measurements of liver size [64]. Hepatomegaly and variants of shape have a wide range of congenital and acquired underlying diseases and are beyond this review on incidentally found focal liver lesions; we refer also to the European Federation of Societies for Ultrasound in Medicine and Biology (EFSUMB) ultrasound textbook [65].

### 3.2. Inhomogeneous Liver Parenchyma

Liver parenchyma is usually characterized by a homogeneous echogenicity slightly more than the right kidney and slightly less than the spleen. An inhomogeneous parenchymal echotexture, without clearly defined focal lesions, may also be regularly encountered on US [66]. Whilst this can represent an incidental normal variant, it may also indicate the presence of underlying disease, including steatosis, congestion, infection, and granulomatous disease. In patients with diffuse liver diseases the laboratory examinations are the basis for diagnosis. The use of elastography [67,68,69,70,71,72,73,74] and quantification of steatosis [75] are prerequisites for hepatic staging regarding fibrosis and cirrhosis and quantification of fat. Biopsy, histological and immunohistochemical examination are the final elements for complete diagnosis.

Nodular regenerative hyperplasia is rare in pediatric patients and almost always diagnosed in association with an underlying disease such as lymphoproliferative and myeloproliferative syndromes, systemic lupus erythematosus (SLE) and the use of medication. NRH is represented by solitary but also, more often and typically, multiple isoechoic nodules. The CEUS pattern is unspecific with isoenhancement in all phases compared to adjacent liver parenchyma [31,76,77].

### 3.3. Elastography

Liver elastography is a valuable complementary US technique, which should be used to further stage the degree of liver fibrosis in patients who may have clinical suspicion of chronic liver disease [70,71,73,74,78]. Point shear wave elastography (pSWE) and/or 2D shear wave elastography (2D-SWE) are available in almost all high-end ultrasound machines, and transient elastography is a stand-alone alternative [67,68,71,73,74,79,80]. The techniques can be applied for pediatric patients [70,78,81]. Liver elastography should be used in all patients before characterizing FLL, since the pretest probability of final diagnoses is different in pediatric patients with and without liver cirrhosis. In addition to elastography, quantification of steatosis has been introduced to further identify patients at risk [75]. The detection of B-mode findings of liver cirrhosis should prompt laboratory investigation. Elastography can be applied before and after CEUS [82].

#### Diagnostic Work-Up and Follow-Up Strategy

When inhomogeneous liver parenchyma is identified, elastography is recommend in order to stage the grade of fibrosis and CEUS is recommended to improve the detection of any focal abnormalities [15]. Follow-up will be determined by the result of CEUS and other clinical findings including laboratory results. Serological laboratory tests and, in some patients, ultrasound guided liver biopsy with histological evaluation, including immunochemistry, is necessary.

### 3.4. Liver Cysts

Epithelial, otherwise called simple liver cysts, are the most common incidental focal liver lesions. Cysts may be broadly categorized as congenital, epithelial cysts, pseudocysts, cystic masses secondary to infection/inflammation or cystic neoplasia. Cysts are observed in 5–20% of routine abdominal US examinations in the entire population [51,83,84], whereas the detection rate in children depending on age are much rarer than reported for adults [13,85].

Liver cysts, as well as other cysts, are characteristically, on conventional B-mode US, round or ovoid, anechoic (echo free), smoothly delineated structures with refraction shadows at their edges, a strong posterior wall echo and post-cystic enhancement resulting from an intensity difference between the beam intensity deep to the cysts and in the cysts and no vascular flow signals on color Doppler imaging. Cysts displaying all these sonographic signs are defined as typical, whereas cysts showing only some of the signs are defined as atypical. Blood vessels mimicking a simple cyst have to be excluded by color Doppler techniques in order to rule out hereditary angiectasia, pseudoaneurysm and other arterioportal venous malformations, including hepatic manifestation of Osler’s disease.

#### Diagnostic Work-Up and Follow-Up Strategy

Typical (“simple”) cysts as an incidental finding do not require further investigation or follow-up. Symptomatic cysts should be further investigated together with those with atypical features, including internal echogenicity, mural nodules, multi-septation and septal calcifications. Absence of enhancement on CEUS is reassuring because it indicates that the cyst is an incidental benign finding. Worrisome features on CEUS include, most importantly, contrast enhancing solid mural nodule(s) and thickened contrast enhancing wall with irregular contour [76,86,87]. The most important features of neoplasia are mural nodules, a wall, and/or septal enhancement on contrast imaging. Differential diagnosis includes abscess formation, parasitic origin, very rare primary malignancies in childhood and cystic hepatic metastases [87,88].

The evolution of hepatic and extrahepatic parasitic infections with, e.g., echinococcus granulosus (hydatid disease) and multilocularis has been recently described in detail and is valid for pediatric patients as well [89,90,91,92].

In conclusion, follow-up is not usually required for incidentally detected liver cysts that are sonographically simple or cysts showing no enhancement on CEUS. CEUS allows prompt assessment of atypical cystic lesions at the time of diagnosis, with improved resolution compared to CT and the advantage of an immediate definitive diagnosis in comparison to contrast enhanced MRI, which needs further resources [15]. MR imaging is the primary modality for further and preoperative evaluation of suspicious cysts. Contrast enhanced CT should be avoided in pediatric patients due to its radiation and limited performance in characterization of cystic lesions.

## 4. Solid Focal Liver Lesions: Introduction

Solid FLL are less common in children than in adults, are usually asymptomatic and present as IFs on imaging studies. Benign lesions are much more common than incidentally detected malignant lesions, depending on the age of the pediatric patients. Hemangiomas are the most common benign tumors of the liver.

On US, FLL other than cysts can be divided into those that are hyperechoic in comparison to the surrounding liver parenchyma, isoechoic and hypoechoic. Most incidentally detected FLL are hyperechoic and represent hepatic hemangiomas which are usually solitary (but in one-third multiple), well defined and vascular or avascular in color Doppler examinations, depending on the age of the patient and, therefore, the respective hemangioma entity. Isoechoic FLL are typically hepatocellular adenoma and focal nodular hyperplasia. Hypoechoic FLL represent a wide variety of etiology. It is generally accepted that most hemangiomas in newborn are hypoechoic. It is a personal observation that within the first year the majority become hyperechoic. Conventional imaging with contrast agents has a limited ability to characterize FLL, and the appearances of benign and malignant masses frequently overlap.

### 4.1. Diagnostic Work-Up and Follow-Up Strategy

Analyzing the pretest probability and correlation with laboratory tests, clinical examination and previous imaging is the first action to be taken and is essential for accurate patient management. The second measurement is contrast-enhanced imaging. The liver is ideally suited for contrast-enhanced imaging such as CEUS due to its dual blood supply and high vascularity.

The use of CEUS for identification and characterization of solid liver masses is recommended by guidelines [15,93]. CEUS is recommended by the NICE guidelines for the characterization of an incidentally detected focal liver lesion in adults whenever unenhanced US is non-diagnostic and further imaging is advised.

Malignant liver lesions typically show variable arterial enhancement and parenchymal phase microbubble washout. Benign lesions typically either show no contrast enhancement in any phase or retain microbubbles in the late phase representing iso- or hyper-enhancement.

### 4.2. Liver Calcification

Localized or multilocular (“starry sky”) liver calcifications are most often harmless and are almost invariably observed as IFs. Focal calcification may be seen as the result of post-inflammatory and often infectious (granulomatous) diseases or region-dependent parasitic bile duct infections. In some cases, the pathogenesis of the calcification remains uncertain. Calcification is characterized as hyperechoic structures which normally show acoustic shadowing distally owing to the reflection and attenuation of the ultrasound. Calcification of cysts are discussed above.

#### Diagnostic Work-Up and Follow-Up Strategy

Liver calcification without an associated mass does not require follow-up.

### 4.3. Vascular Malformations, Vascular Tumors and Hemangioma

Vascular tumors and vascular malformations are best categorized according to the 2018 International Society for the Study of Vascular Anomalies (ISSVA) classification, which incorporates the biology and genetics of vascular malformations in its nomenclature [94]. The two most common vascular malformations in infants and young children are congenital hemangioma (also called solitary hemangioma) and infantile hemangiomas (also called multifocal or diffuse hemangiomas), both of which are vascular tumors (Figure 1). Although these previously were referred to as “hemangioendothelioma,” this term is now most appropriately used to describe locally aggressive or borderline malignant vascular tumors, such as the hepatic epithelioid hemangioendothelioma [94]. Vascular malformations, previously called “hemangiomas” and still commonly called this in much of the literature, are seen after puberty. Vascular malformations are best named for the type of vessel present or flow rate of the lesion (i.e., fast flow or slow flow), and in the liver, slow flow malformations are by far most common [95,96].

#### 4.3.1. Hepatic Hemangioma

Hepatic hemangiomas are the most common benign solid FLL in pediatric patients, and they are observed more often in girls than in boys [76,83]. The frequency in adults (up to 7%) is much higher compared to children [83]. Explanations for the differences in reported frequencies include growth during hormone-induced puberty. Liver hemangiomas can be solitary but may also be multifocal and, in very rare cases (in childhood), diffuse. An association with focal nodular hyperplasia (FNH) is relatively common [97,98,99]. Congenital and infantile hemangioma have been differentiated according to histological features and biological behavior [100,101,102,103,104,105,106] and imaging features [104,105,107,108,109]. Histological terms such as capillary or cavernous hemangioma should be avoided by analyzing imaging.

Most hemangiomas demonstrate typical sonomorphological features in conventional B-mode ultrasound and Doppler imaging. The conventional B-mode ultrasound criteria of typical hemangioma are as follows: less than 3 cm in diameter, next to a liver vein, hyperechoic in comparison to the surrounding liver parenchyma, homogeneous echogenicity, round or slightly oval shape, well-defined with a smooth outline, lobulated, feeding and draining vessels, absence of any signs of invasive growth, absence of any halo sign and posterior acoustic enhancement owing to blood filled capillaries.

Although hemangiomas are highly vascularized masses, from a histopathological perspective, they essentially consist of a large number of capillary-sized vessels. Conventional color Doppler ultrasound often detects little or no blood flow inside the haemangioma due to slow flow. The supplying and draining vessels (“feeding vessels”) may be visualized (depending on the ultrasound system’s performance) at the edge of the lesion.

CEUS features of hemangioma may be divided according to size and enhancement features into at least six types: The typical and small hemangioma, ≤30 mm, peripheral nodular contrast enhancement and complete or incomplete centripetal fill-in [44,76,108]. The typical and larger hemangioma, >3 cm–10 cm, peripheral nodular contrast enhancement and complete or incomplete centripetal fill-in with or without regressive changes. Larger lesions may appear heterogenous due to calcification, necrosis and haemorrhagic component. The giant hemangioma (>10 cm), peripheral nodular contrast enhancement and complete or incomplete centripetal fill-in with or without regressive changes. The shunt hemangioma with complete fill in within a second with or without surrounding focal fatty sparing. The atypical hemangioma totally thrombosed with regressive changes and neonatal and infantile forms of hemangioma.

Infantile hepatic haemangiomas constitute vascular malformations. These can be focal, multifocal or diffuse, with the multifocal form being associated with cutaneous and other organ manifestations of hemangiomas. The Kasabach –Merritt syndrome (KMS) is an extreme form with life-threatening consumptive coagulopathy in the presence of a rapidly enlarging vascular tumor(s) of the liver. It usually presents in early infancy, but onset in the early neonatal period has been reported. The association with vincristine use has been described [83]. Hemangiomas generally do not require surgery with very few exceptions, including obvious and severe symptoms with diagnostic uncertainty [88].

Important differential diagnosis with sometimes confusing contrast features includes peliosis [110] and other forms of degenerative metamorphosis. Hemangioendothelioma [13,111,112] and angiosarcoma [113] are characterized by washout in the portal venous and late phase and require, therefore, biopsy.

Hemangioma will only require treatment if the patient develops significant and long-term symptoms. Life-threatening syndromes require special pediatric knowledge and should be treated in specialized centers.

#### 4.3.2. Congenital Hepatic Hemangioma

Congenital hepatic hemangioma (CHH) is a rare vascular tumor that develops in utero. The tumor is present at birth, is typically largest at birth, and is usually diagnosed within six months of life. We refer to the published literature [101,114,115,116].

#### 4.3.3. Infantile Hepatic Hemangioma

Infantile hepatic hemangiomas are vascular tumors that develop in the first weeks or months of life, although a small precursor lesion may be seen. Infantile hemangiomas characteristically have rapid growth within the first year of life followed by gradual involution over the course of several years, with a small amount of persistent fibrofatty tissue. Infantile hepatic hemangiomas are multiple or innumerable/diffuse and are associated with cutaneous infantile hemangiomas, which are histologically identical tumors with endothelial glucose transporter 1 (GLUT1) positivity immunohistochemically. Clinically, a patient may have complications related to vascular shunting, including high output heart failure and hypothyroidism, because these tumors express type 3-iodothyronine deiodinase [101]. If treatment of infantile hepatic hemangiomas is necessary, propranolol is the treatment of choice [101].

CEUS may not be necessary for diagnosis when multiple or innumerable liver lesions are encountered at ultrasound in a patient with multiple cutaneous infantile hemangiomas. As grayscale imaging, infantile hemangiomas are typically more homogeneous in appearance than congenital hemangiomas. If a patient presents with liver lesions prior to identification of cutaneous infantile hemangiomas, the primary differential diagnosis consideration is liver metastases and CEUS is helpful for differentiation of the two entities. At CEUS, infantile hemangiomas typically have peripheral hyperenhancement with very rapid homogeneous centripetal fill-in [101]. There is usually sustained enhancement in the portal venous phase with hyperenhancement, isoenhancement and mild washout all possible in the delayed phase [114]. Importantly, there is no early or marked washout, which helps distinguish infantile hemangiomas from metastases.

### 4.4. Vascular Malformations Other than Hemangioma

As detailed above, hemangioma is a term used ubiquitously in the literature, but is a distinct, non-tumoral malformation, different from both congenital hemangioma and infantile hemangiomas [117]. These lesions are more appropriately called vascular malformations and are encountered in an adolescent or adult, not an infant or young child [94]. Vascular malformations may be further characterized as slow or fast flow depending on which vessels are present and are more common than the true hemangiomas encountered in infants and young children. In the liver, venous malformations are the most common vascular malformation [96].

The appearance at CEUS is similar to multiphase CT and MRI, with peripheral discontinuous nodular enhancement and expanding puddles of contrast material that fill-in from outward to inward. Fill-in is usually complete, but partial fill-in may be seen due to central clot or fibrosis. Most commonly, there is no contrast washout, although mild late washout has been reported. The presence of washout should not be mistaken for malignancy in these lesions if the characteristic arterial phase enhancement pattern is present [118].

#### Focal Nodular Hyperplasia (FNH)

FNH is the second most frequently observed FLL in pediatric patients [13,83,119] accounting for about 2–4% of liver tumors [13,76,85,93,120,121]. FNH, and its important differential diagnosis of HCA, are two benign, mostly incidental, hepatic FLL which occur predominantly in post-puberty girls and young women. It is of importance that FNH is not a true neoplasia but is instead a regenerative lesion characterized by hyperplasia of hepatocytes, malformed bile ducts and vascular malformation with atypical portal vein branches [43,83,122].

Similar to hemangioma, multiplicity has been reported in about one-third of patients and association with hemangioma is obvious [43,123,124].

FNH is typically an isoechoic or slightly hypoechoic compared with the surrounding liver parenchyma and hypervascular FLL of variable size, with age and size dependent regressive changes. FNH are difficult to detect if isoechoic (isodense, isointense) and if small [125].

The vascular tree can be centrally located with centrifugal filling pattern (1), eccentric at the border of the lesion with respective filling pattern (2), with multiple supplying arteries at different sites (3) and without typical vessel architecture by imaging methods (4) (Figure 2). FNH in older patients (>35 years old) may shrink and vanish. The unique real-time features of CEUS, including destruction of microbubbles and replenishment, are particularly useful for the visualization of the vascular supply in FNH [55,126].

Parametric ultrasound techniques are helpful to illustrate the enhancement pattern [55,126,127,128]. The so-called wheel-spoke (spoke-wheel, stellate) phenomenon is typical for FNH but interobserver reliability may be lower than expected and FNH can be confused with HCA and HCC in the non-cirrhotic liver. FNH typically show hyperenhancement in the portal venous and late phase (>95%) [129], which is the distinctive criteria to differentiate HCA from FNH. HCA do not contain portal vein branches, therefore, typically show washout in the portal venous and late phases [15,43,130]. A low mechanical index should be used to avoid bubble destruction, which may mimic washout in an otherwise hypervascular FLL [58,59,131].

The central scar is histologically present in only 70% of FNH [132] and, therefore, neither sensitive nor specific [123,132,133,134] since other larger FLL, including hemangioma and HCA, also show centrally located regressive changes [43,130]. Contrast enhanced MRI may be used in uncertain cases, whereas contrast enhanced computed tomography should not be used to characterize probable benign FLL in pediatric patients due to its significant radiation exposure and lower accuracy [45,76,86,122]. Biopsy and histological evaluation are nowadays rarely necessary but should be considered in patients with uncertain features and patients with underlying malignant diseases.

FNH will only require treatment if there are significant and long-term symptoms [83].

### 4.5. Hepatocellular Adenoma (HCA)

Hepatocellular adenoma is a benign neoplasm that more frequently occurs in females than males and is typically seen after the first decade of life. Understanding of these tumors continues to improve with increasing molecular and genetic knowledge. Adenomas are currently categorized into seven distinct subtypes: inflammatory; HNF-1α-inactivated; β-catenin-activated (also called β-catenin mutated, caused by a mutation of exon 3); weak β-catenin-activated (caused by a mutation of exon 7/8); sonic hedgehog pathway activated; and unclassified hepatocellular adenoma; as well as two subtypes with overlapping features of the inflammatory and β-catenin-activated subtypes [115,135]. Many adenoma subtypes are not yet well-described at CEUS. The few that have are further detailed below.

HCA is a rare FLL. The relation of FNH/HCA can be estimated to be ten to twentyfold higher in children than HCA in adult patients. In pediatric patients, HCAs are relatively frequently seen in patients with glycogen storage disease (GSD 1). HCAs are also more commonly associated with diabetes mellitus and obesity (metabolic syndrome) in pediatric patients [136]. Other influencing factors include, mainly in adult but theoretically also in pediatric patients, high alcohol intake, oral contraceptives or anabolic steroids [83,85,137]. If more than five hepatic adenomas are seen in a patient, the term “hepatic adenomatosis” can be used [137].

Histologically different subtypes of HCA have been described based on molecular (mutation present or not), pathologic features (steatotic, inflammatory) and immunohistochemistry [83,138,139,140].

Hepatocyte nuclear factor 1a-inactivated HCA (most common according to Zucman-Rossi) [140].

*β*-catenin-activated HCA, inflammatory HCA (steatotic or non-steatotic) and HCA with genetically determined underlying diseases, e.g., glycogen storage diseases are non-classified.

There are no typical criteria in B-mode ultrasound. In a B-mode ultrasound of an otherwise normal liver, HCAs are roundish, well-circumscribed and usually isoechogenic with the surrounding liver tissue since HCAs are constituted by hepatocytes. Owing to this lack of echogenicity, HCA can be very difficult to differentiate from the surrounding liver tissue and small HCAs have been rarely described [137]. In steatosis, HCAs may be poorly echogenic, whereas in patients with storage diseases (e.g., glycogenosis), HCAs can be hyperechoic due to lipid content and/or glycogen. Larger HCAs > 50 mm show regressive changes with changes of echotextures.

Color Doppler techniques can visualize increased peripherally predominant internal arterial hypervascularity [76,83,125]. However, this vascular pattern can also be encountered in HCC and hyperperfused metastases and is therefore not pathognomonic. Calcification and other regressive changes can be observed depending on the size of the lesion in the same frequency as in large FNH.

The characteristic CEUS pattern of enhancement shows quick and intense homogenous peripherally predominant arterial phase hyperenhancement (APHE) and centripetal distribution. Teleangiectatic HCA may show more centrifugal enhancement patterns.

Based on the exact histologic type of HA, washout can be typically observed [122,137] and differentiation from HCC is difficult, often requiring tissue diagnosis or surgical resection [43,110,141,142,143]. If the contrast agents Levovist^®^ [130] or Sonazoid^®^ [45] are used, the HCA also appears iso- or hyper-enhancing in late phases due to sinusoidal-derived mechanisms, such as Kupffer cell phagocytosis.

Differentiation is essential because of the different therapeutic approaches; HCA is an indication for surgery at least where growth has been demonstrated on serial imaging in HCA > 50 mm because of the risk of hemorrhage, rupture and potential malignant transformation due to *β*-catenin lesions demonstrating genetic overlap with HCC. In contrast, FNH can be managed conservatively in most patients. Small HCA might be closely followed up with the removal of potential precipitating factors. In doubtful cases, a multidisciplinary approach has been advocated [83].

The molecular classification of HCA is of prognostic relevance; therefore, histological evaluation is mandatory. In pediatric patients, the underlying predisposition is also of importance. Biopsy specimens should be taken from the surrounding parenchyma as well in order to determine underlying diseases.

### 4.6. Inflammatory Hepatocellular Adenoma

Inflammatory hepatocellular adenoma (previously called telangiectatic adenoma or telangiectatic FNH) is the most common subtype and exposes the highest risk of spontaneous rupture, hemorrhage and necrosis in larger adenoma [115,144,145,146]. Rarely observed in childhood, this type of adenoma typically shows mainly peripherally predominant arterial phase hyperenhancement and wash-out has been observed [146,147,148], therefore, biopsy may be necessary for diagnosis.

### 4.7. HNF-1α-Inactivated Hepatocellular Adenoma

HNF-1α-inactivated hepatocellular adenomas (previously called steatotic adenomas) account for 35–50% of adenomas and are usually seen in females with oral contraceptive use [145] and in patients with autosomal dominant maturity onset diabetes of the young, type 3 (MODY3) [144]. Treatment includes discontinuation of oral contraceptive use [115,145]. HNF-1α-inactivated hepatocellular adenomas have the lowest potential for bleeding and the lowest risk of malignant transformation [135].

At CEUS, HNF-1α-inactivated hepatocellular adenomas are homogeneously hyperenhancing in the arterial phase. Lesions show hyper- or iso-enhancement in the portal venous and late phases, with hypoenhancement less common [146,147,148].

### 4.8. β-Catenin-Activated Hepatocellular Adenoma

β-catenin-activated hepatocellular adenomas account for 10–18% of adenomas. This is the subtype most frequently found in males and there is an association with exogeneous androgen exposure, such as for muscle building or for treatment of Fanconi anemia. This subtype also occurs in patients with familiar adenomatous polyposis and glycogen storage disease [115,144,145]. This adenoma subtype has the highest rate of malignant transformation at about 50% [115,144].

At CEUS, arterial phase hyperenhancement is typically diffuse and homogeneous, with either portal venous or delayed phase washout in almost 90% of lesions. Given overlap in appearance with HCC, biopsy may be necessary for diagnosis.

### 4.9. Von Meyenburg Complex

Biliary microhamartomas (von Meyenburg complex, VMC) are ductal plate malformations characterized by very small cystic dilatations of peripherally located intrahepatic bile ducts lying within fibrous stroma. They may be single but are typically, and much more often, multiple. VMC may have different imaging appearances depending on their histopathology and size. Other cysts and cystadenoma have to be differentiated (Figure 3).

The US appearance of VMC is specific with a coarse heterogenous appearance of the liver periphery, while innumerable micro-nodules (2–10 mm) of either decreased or increased echogenicity may be observed [149]. High frequency transducers are a prerequisite to examine the periphery of the liver with up 5% of the detection rate (personal unpublished data). In more detail, biliary hamartomas are described as coarsely heterogenous hyperechoic (when mostly solid), of mixed echogenicity with cystic components or purely anechoic (cystic). In a fatty liver, hamartoma may be hypoechoic. In most reported cases, hamartomas are small (5 mm in diameter) hyperechoic lesions, and they can change to hypoechoic or microcystic lesions over time, as they become larger, demonstrating small-cystic echotexture of the liver parenchyma. Particularly, the microcysts and their biliary content are at an early stage so small that ultrasound can reverberate within them, causing the “comet tail” artefact and, thus, requiring differentiation from aerobilia and very small intraductal bile stones. In terms of size, VMC nodules range from 5 to 10 mm [149].

CEUS findings have been reported in a few histologically proven patients with underlying malignant diseases with isoenhancement in the late phase using Levovist [130]. The association with intrahepatic cholangiocarcinoma in the setting of ductal plate malformation has been challenged. Long-term imaging and laboratory follow-up may be advised in specific cases but more often no follow-up is required.

### 4.10. Mesenchymal Hamartoma of the Liver

Mesenchymal hamartoma of the liver is a heterogenous, and often multi-cystic, mass typically seen in the first two years of life.

In B-mode, ultrasound hamartomas may have different appearances depending on their histopathology and size. They may be single or multiple [150]. Larger hamartomas in the first two years of life are described as hyperechoic (when mostly solid) or presenting mixed echogenicity with cystic components or be purely anechoic (cystic). Calcification and hemorrhage are rarely seen. In a fatty liver, they can be hypoechoic. Conventional US will demonstrate a large mass with multiple cystic spaces and solid components.

In most reported cases, hamartomas are small (less then 5 mm in diameter) hyperechoic lesions, some presenting the “comet tail” artefact [151,152]. Some authors may use the term identical to VMC. However, in children, where they are usually large and fast growing lesions, they are easily detected [153]. In most cases, hamartomas do not exhibit any flow signal, but in rare cases one vessel may be spotted. Large lesions often displace surrounding structures, thus modelling of the liver parenchymal vessels may be visible [154,155]. In fast growing hamartomas, exceeding 10 cm in diameter, disturbance of venous flow is observed, including signs of portal hypertension [153].

There is no consistent CEUS enhancement pattern. Some cystic parts of the lesions show no enhancement in either of the vascular phases due to their cystic composition, some show enhancement such as surrounding liver parenchyma, and occasionally some of them show bright and early enhancement in the arterial phase with a gradual decrease in intensity that is consistent with liver parenchyma [125,151,155]. Feeding vessels may be prominent [152]. Concerning the pattern of hamartoma, enhancement malignancy may be excluded in most cases.

Because of their non-specific image appearance, they can mimic a wide range of focal liver lesions from haemangiomas, cysts to metastases or HCC in cirrhotic liver.

### 4.11. Cholangiocellular and Bile Duct Adenoma

Cholangiocellular adenomas are frequently small neoplasia appearing as small solid hypoechoic nodules, typically in a subcapsular location [151,156]. These lesions typically show strong arterial phase hyperenhancement on CEUS and washout due to lack of portal vein branches [156]. Other authors used the term bile duct adenoma for a heterogenous group of FLL [130,154,156].

## 5. Focal Fatty Infiltration and Focal Fatty Sparing (Focal Fatty Lesions, FFL)

Today, non-alcoholic fatty liver disease (NAFLD) is also commonly seen in the pediatric population and has been described as “epidemic obesity” [70,157]. Fatty infiltration (steatosis) was generally considered to be a diffuse process involving the entire liver. FFL are typically observed in adult patients and have been identified in about 50% at autopsy and with different frequency of up to one-third in imaging studies [11,12,14,53,65,158,159,160,161,162,163]. FFL can be divided into hyperechoic focal fatty infiltration (FFI) and hypoechoic focal fatty sparing (FFS). The frequency in children is much lower and depends on the weight of the children examined. Both FFI and FFS can be identified in the liver hilum and in the liver periphery. Both types of FFL do not usually cause any mass effect or displace vessels [160]. The underlying mechanism has been identified in specific vascular supply and also in biopsy proven cases and autopsy studies [158,159]. The location of these entities is a helpful diagnostic feature, since they are commonly located in the liver hilum, adjacent to the porta hepatis or the falciform ligament. Other more diffuse forms include the geographic, segmental and perivascular forms, that surround hepatic or portal venous branches [164]. Focal and diffuse fatty abnormalities are commonly seen as incidental findings on routine US [164]. Severe diffuse fatty infiltration can markedly reduce the US beam attenuation and reduce the ability to detect any focal liver lesion and should be overcome by improved machine settings [162].

### 5.1. Focal Fatty Sparing (FFS)

In the majority of patients with hepatic steatosis, distinctive hypoechoic areas in the liver hilum can be detected and should not be confused with a mass lesion. FFS typically shows centrally located feeding arterial vessels with less insulin and fat concentration, thus demonstrating the underlying pathological process of different vascularization of the liver hilum. The ultrasound-derived fat fraction (UDFF) is lower than in the surrounding steatosis and the stiffness measured by shear wave elastography is slightly higher [67,68,69,75]. Shunt hemangioma are often observed in FFS [15,134].

### 5.2. Focal Fatty Infiltration (FFI)

FFI can often be displayed in the liver hilum [159] and next to the hepatic teres ligament [160]. In FFI, changes in arterioportal venous perfusion have been suggested as the pathophysiological explanation (with predominant portal venous flow and high content of insulin and fat). Haemangiomas may mimic such lesions; however, not all bright lesions in the liver are haemangiomas. FFI occur in about 40% of patients taking corticosteroid medication. Different-sized vacuoles or different types of lipid deposition are the likely cause, because a change in appearance over time can be observed. The ultrasound-derived fat fraction (UDFF) is higher than in the surrounding liver parenchyma [75].

CEUS examination is “the” useful problem solving tool for the characterization of a focal fatty abnormality since it is prompt and readily available, thus demonstrating the underlying pathophysiological mechanism with demonstration of feeding vessels and almost identical arterial phase enhancement when compared with the surrounding liver parenchyma and no washout in the portal venous and late phases [161]. CT and MR imaging is generally not indicated and should be reserved for high-risk patients with underlying malignant diseases, scientific questions, and aims to determine non-invasively the hepatic fat content [45,69,75,76,162,165,166].

### 5.3. Inflammatory Focal Liver Lesions

Inflammatory focal liver lesions are rarely asymptomatic and, therefore, rarely incidental findings. Phlegmonous infiltration and abscess formation will not be covered here. Autoimmune diseases typically show FLL in symptomatic patients. Examples include the inflammatory pseudotumor [167] and granulomatous diseases, e.g., sarcoidosis [168,169]. CEUS features have rarely been observed in children. According to our own experience, washout can be typically observed in larger FLL > 20 mm.

## 6. Malignant Focal Liver Lesions

Malignant focal liver lesions are rarely asymptomatic and, therefore, rarely incidental findings.

The most common primary malignant tumor among children < five years is hepatoblastoma, and later (15–19 years old) hepatocellular carcinoma [125,170,171]. Other malignant focal liver lesions in pediatric patients include primary (cholangiocellular carcinoma, sarcoma and lymphoma) and secondary liver tumors (metastases) and will be discussed in more detail in a separate paper. In pediatric patients with healthy liver parenchyma and normal portal venous predominant perfusion, late and post vascular phase enhancement provides important information regarding the character of the lesion: most malignant lesions are hypo-enhancing, showing washout, while the majority of solid benign lesions are iso- or hyper-enhancing [15,16,17,18]. The arterial phase varies according to entities with rim-enhancement as a sign of malignant behavior. Analyzing the arterial phase does not allow differentiation between benign and malignant FLL, which is true for all imaging methods.

CEUS shows which pediatric patients needs further investigations by detecting washout in the portal venous and late phases. The further investigations include biopsy and histological examination, computed tomography (CT) for staging purposes and magnetic resonance imaging (MRI). Important laboratory biomarkers to support imaging have to be taken into account [172].

### 6.1. Hepatoblastoma

Hepatoblastoma is by far the most common hepatic malignancy in children younger than five-years of age (90%) [171] and accounts for around 60% of pediatric malignancies in general. The leading symptoms are abdominal distension and weight loss [116].

Much more commonly, hepatoblastoma appears as a larger solid mass, often multilocular, that is size dependent with pseudocystic areas, necrosis and calcifications. The diagnosis is assisted by a typically marked increase in alpha-fetoprotein. The oncological staging methods are primarily computed tomography and magnetic resonance imaging [173]. CEUS typically shows arterial phase hyperenhancement and early and punched washout [76,170]. The role of CEUS in hepatoblastoma, especially during follow-up, has not been clearly defined.

The important differential diagnosis is hepatocellular carcinoma.

### 6.2. Hepatocellular Carcinoma

In pediatric patients, HCC is strongly associated with underlying pathology and, therefore, most often not an incidental finding. Patients with vertical transmitted chronic viral hepatitis B, with or without liver cirrhosis and congenital metabolic disorders such as glycogen storage disease, tyrosinemia and longstanding hepatic vein obstruction [174,175,176], and many other diseases put patients at risk of HCC. It is of importance that the viruses hepatitis B and C can be effectively treated. The role of elastography is discussed above. The development of HCC occurs in a well-established continuum starting from a regenerative nodule and proceeding to a dysplastic nodule and then to a well and low-differentiated HCC, all before the development of a poorly differentiated HCC [177]. HCC in non-cirrhotic liver in patients with chronic virus hepatitis B [142,143], fibrolamellar HCC [178] and mixed forms [179] are relatively more often diagnosed in children compared to adults.

The work-up for suspected HCC in liver cirrhosis has been described in detail, and the imaging criteria do not really differ in children compared to adults [142,143,178,180]. With the introduction of CEUS LI-RADS criteria, the use of CEUS has become more standardized and reproducible [46,181,182,183,184,185,186,187]. The LI-RADS imaging hallmarks of HCC are the size, the arterial phase hyperenhancement (APHE) and the portal venous and delayed phase washout. The most important CEUS feature is APHE [188,189]. Evidence exists regarding the CEUS appearances of HCC in children with CEUS showing hyperenhancement in the arterial phase with mild and later washout compared to hepatoblastoma [173,190,191,192]. Liver Imaging Reporting and Data System (version 2018) had moderate sensitivity but low specificity for the diagnosis of pediatric hepatocellular carcinoma (HCC), which had low frequencies of the major criteria used for adult HCC diagnosis [191,192].

CEUS can also be used to characterize the very rare intrahepatic cholangiocarcinoma (ICC) in pediatric patients with specific diagnostic criteria being included in the CEUS LI-RADS classification system. The LR-M category for metastases and ICC in liver cirrhosis, including peripheral rim APHE, marked washout earlier than 60 s [46,181,182,183,184,185,186,187].

### 6.3. Primary Sarcoma of the Liver

Primary sarcomas of the liver are very rare and most often large and symptomatic at time of diagnosis. Conventional imaging shows often palpable large and heterogeneous FLL due to intralesional hemorrhage, necrosis and cystic degeneration. In such FLL, e.g., embryonal sarcoma [193,194,195], rhabdomyosarcoma [196,197] and leiomyosarcoma the arterial phase is non-specific. The so-called rim sign, and more importantly, in the portal venous and late phases washout and hypoenhancement indicate malignancy. The lack of more specific imaging findings will raise the need for tissue diagnosis after surgical resection or biopsy. A variety of such tumors have been documented in the EFSUMB European Pediatric Registry [16].

### 6.4. Primary Hepatic Lymphoma

Primary hepatic lymphoma (PHL) is a very rare malignant tumor of liver, occurring in <0.1% of all non-Hodgkin’s lymphoma (NHL) [198,199]. Diffuse large B-cell lymphomas (DLCL) and mucosa-associated B-cell lymphoma (MALT) are considered the two most common types of PHL [199]. PHL may be solitary or showing multiple hypoechoic or isoechoic lesions, sometimes diffusely infiltrating [200,201]. The diffuse pattern is usually seen in secondary involvement of the organ or in post-transplant lymphoproliferative disease. CEUS typically reveals APHE and early and punched washout [202,203,204]. Similar CECT and CEMRI features have been described as well [205,206,207,208,209,210,211]. The diagnosis is finally achieved by biopsy and histological examination, including immunohistochemistry. Mucosa-associated lymphoid tissue (MALT) lymphoma is a low-grade malignant B-cell lymphoma that was first described by Isaacson and Wright in 1983 [212]. Whereas the most common site of MALT lymphoma is the stomach, MALT lymphomas can arise at any extranodal site and also as primary hepatic MALT lymphoma [213,214,215]. Although the etiology of primary hepatic MALT lymphoma remains unknown, most reported cases have implicated chronic inflammatory liver diseases, including hepatitis B or C virus but also EBV, HIV-infections, steatohepatitis, autoimmune hepatitis, liver cirrhosis, systemic lupus erythematosus, or the use of immunosuppressive drugs, [198,199,201,202,204,216,217,218]. Optimal treatment strategies have not been established so far [216,217].

### 6.5. Hepatic Metastases

Hepatic metastases in general represent advanced disease and will rarely be found in a pediatric patient with no symptoms. The most common pediatric hepatic metastases originate from neuroblastoma or Wilm’s tumor. Metastases are often hyperenhancing in a diffuse (if small) or rim pattern (if larger) in the arterial phase, with washout in portal venous phase in almost all patients [42,45,83]. CEUS might be more sensitive than CT or MRI, but the latter are mandatory as the multiplicity of lesions in different organs including the lung cannot be characterized simultaneously with CEUS, and the complete body staging needs to be performed.

## 7. What Is the Impact of an Incidental Finding in Pediatric Ultrasound?

The following issues are almost identical for the evaluation of IF in all organs and will be briefly summarized [2,3,4,5,6,7,8,219].

### 7.1. Psychological and Social Burden

The effects of incidentally discovered pathological findings in a healthy individual can be unpredictable, complex and far-reaching, and may be underestimated by patients and clinicians [7,220,221,222,223,224,225]. Potentially serious incidental findings may provoke anxiety in the individual concerned and will affect the perception of their own health status [2,7,64,66,222,226,227,228,229,230,231]. Due to a lack of evidence, and perhaps also driven by concerns and irrational anxieties, further management of incidentally detected lesions will not always follow reasonable and evidence-based pathways. In addition, patients with different types of personality will respond in different ways to the disclosing of an incidental finding, with about 10% experiencing severe stress [228]. We would expect that patients would prefer to find out the nature of the lesion at the same appointment, rather than being referred for an additional MR imaging or CT scan. The direct communication between the patient and the examiner in US is another advantage over cross-sectional modalities where the patient has no direct contact with the radiologist.

Unnecessary further investigation and (over-)treatment may result in a potentially injurious and expensive cascade of tests and procedures, depending on specific health care organization insurance systems, e.g., Beveridge Model, Bismarck Model, NH Insurance Model, Out-of-pocket payments, et cetera [7,222,223,224,232,233,234]. Furthermore, incidental findings may adversely affect individual future perspectives [235] and medical and life insurance status [2,222].

### 7.2. Benefits and Economic Consequences (Costs)

IF may also require costly, potentially uncomfortable and sometimes invasive procedures to achieve a confident diagnosis, and can result in long-term follow-up. Further investigating an incidental finding should be organized in such a way that no pathology is missed, characterization is precise, but also the cost is kept to a balanced minimum, which is a general challenge for healthcare systems [9,236,237,238]. CEUS allows faster diagnosis compared to CECT and CEMRI and, therefore, is time saving [239]. CEUS can be performed at the same time as the initial examination, thus avoiding further imaging in the majority of pediatric patients [16,17,240]. CEUS examination in pediatric patients is at least equally sensitive and specific (accurate) for the differentiation of benign from malignant FLL [16] compared with CT and MR imaging [15], and it is more cost-effective [241]. The sensitivity and specificity were reported with 96% and 97%, respectively [242]. Cost saving of CEUS is more obvious when comparing CEUS with MRI and CT [239,242]. The reported findings suggest a role for targeted educational efforts, collaborative partnerships, and other initiatives to foster greater adherence to written recommendations, such as standardized classification of IFs and their consequences, as well as stronger language within reports when no follow-up testing is recommended [231,243,244].

In conclusion, CEUS is an attractive alternative to CT and MR imaging and, most importantly, is accurate, avoiding radiation exposure and saving costs [16,17,239,240,242,245,246].

## 8. Strategy

### 8.1. Iso- and Hyperechoic FLL

Most hemangiomas are “typical” and <30 mm and can be correctly identified by conventional B-mode and color Doppler imaging; further measures are not often necessary. Careful history-taking of symptoms, any underlying medical conditions or family history, and reviewing liver function tests, bilirubin and cholestasis indicating enzymes, are essential. The testing of tumor markers such as alpha-feto protein (AFP), carcinoembryonic antigen (CEA) and cancer antigen 19-9 (CA 19-9) is not generally recommended as a screening test.

If there is any doubt CEUS can be promptly used as a problem-solving tool, we found that, in about 30% of patients, there will be an atypical criterion and at least in such cases contrast-enhanced imaging procedures should be used (CEUS, MRI), depending on local protocols, availability and expertise. It should also be mentioned that hemangiomas in patients with fatty livers (hepatic steatosis) can appear to be isoechoic or hypoechoic when compared to hyperechoic parenchyma. Lesions suspected of being hemangiomas which show wash-out in the portal venous phase should be biopsied and histologically confirmed [44,50]. The differential diagnosis includes HCA [137], HCC [142], hemangioendothelioma [111,112], peliosis [110] and metastases especially of neuroendocrine origin [247].

### 8.2. Isoechoic FLL

The most commonly isoechogenic liver lesions are focal nodular hyperplasia and hepatocellular adenoma. Contrast-enhanced imaging and promptly available CEUS are required to exclude malignancy and to determine the diagnosis [43,50,97,98,99,132,133,248]. In contrast to FNH, portal veins and bile ducts are not present in hepatocellular adenoma. Therefore, FNH and HCA can be differentiated through analysis of the portal venous contrast phase, which shows a typical hypoenhancement with HCA < 50 mm, whereas FNH show hyperenhancement in 95% [43,50,130] if bubble destruction is avoided [59].

### 8.3. Hypoechoic FLL

Malignant differential diagnoses must be excluded with a high degree of certainty, and this is only possible using contrast enhanced imaging techniques [50]. Washout in the late phase is a decisive indication for liver biopsy [50,123].

## 9. Inconclusive CT/MRI Findings

There are circumstances in which a CT or MR imaging examination may fail to adequately characterize a focal liver lesion, especially if this is incidentally found in a non-specifically protocolled examination. These include the detection of a liver lesion during a single-phase CT examination for trauma or when investigating the acute abdomen, excessive artifacts (especially motion) degrading image quality in CT and MR IMAGING, suboptimal phase timing acquisition or small size that precludes confident characterization (particularly applicable in CT) [47,48,49]. In these instances, US with the potential addition of CEUS can be used as a problem-solving tool for the characterization of an incidentally found focal liver lesion [249].

## 10. Conclusions

The detection of an incidental FLL during US examination in pediatric patients is a common scenario but much rarer than in adult patients. Many factors need to be taken into consideration, in order to appropriately manage each individual patient and reach an accurate diagnosis. Firstly, carefully assess the pretest probability and, most importantly, include age (newborn, infancy), presentation of the patient, medical and family history (including patient’s background risk of malignant and inflammatory liver disease), laboratory data and liver stiffness using elastography. Secondly, assess the lesion’s characteristics, including size and echogenicity. This combined approach highlights an important concept for the management of every incidentally found FLL by including all clinical information available and a multiparametric imaging approach. Pediatric patients should be subcategorized into no, low and high risk for clinical and imaging features. In addition, the psychological, healthcare and economic impact of every imaging modality should be taken into account for appropriate management, diagnosis and follow-up of these patients.

## 11. Summary Statements

An incidental finding (IF) during an imaging examination is the unintended and unexpected discovery of an abnormality that may be clinically significant (or not).The term IF can be used for any focal changes of the liver echogenicity and architecture that is causing no symptoms to the patient.Liver incidentalomas are frequently detected.Ultrasound is the primary and first-line imaging modality for the evaluation of liver pathology in children, including liver incidentalomas.Liver elastography should be applied in all patients with liver pathology to determine underlying risk constellation.Conventional B-mode ultrasound is accurate for the diagnosis of simple hepatic cysts, typical hemangiomas and hydatid cysts. As a result, such lesions could be followed up with US.Contrast-enhanced ultrasound (CEUS) is appropriate to precisely characterize solitary and multiple solid and cystic FLL with excellent accuracy in differentiating benign from malignant lesions.CEUS has an excellent safety profile.MRI is probably superior for the documentation of multiple lesions prior to surgery.Once characterization with CT/MRI has been initially performed, long-term follow-up can be undertaken with US/CEUS if necessary.Benign hepatic tumors are commonly observed in adults, but more rarely reported in children.The pretest probability—including presentation of the patient, past medical and family history with patient’s background risk of malignant and inflammatory liver disease (“all clinical information available”), laboratory data and liver stiffness using elastography—and the lesion’s characteristics including size and echogenicity are taken into account for appropriate management, final diagnosis and follow-up of these patients.According to the factors mentioned, patients can be subcategorized into no, low and high risk for clinical and imaging features.The psychological, healthcare and economic impact of every imaging modality using a multiparametric imaging approach should be taken into account.In some patients, imaging will be inconclusive, and some form of tissue diagnosis will be required, either imaging-guided biopsy or surgical resection.In doubtful cases, a multi-disciplinary team (MDT) dedicated to liver diseases should be consulted.

## Figures and Tables

**Figure 1 cancers-15-02360-f001:**
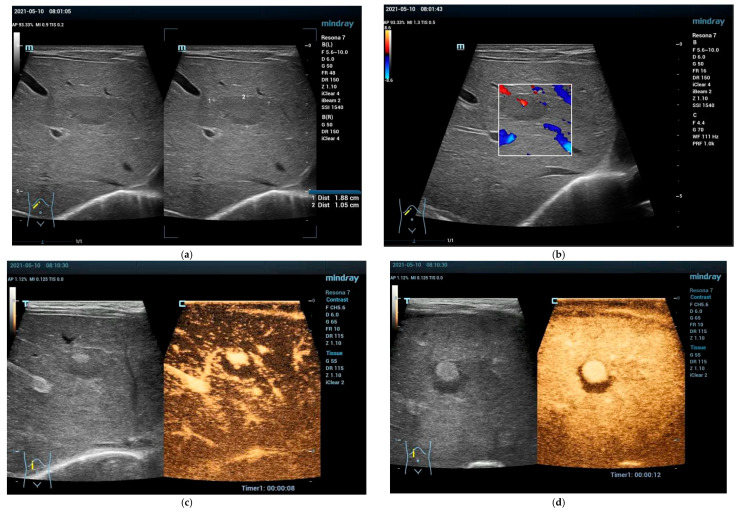
Congenital hemangioma, 6-day-old girl. Incidentally detected FLL. No cutaneous hemangiomas or other malformations visible, no symptoms. B-mode with 19 mm × 11 mm hypoechoic FLL in segment 4 (**a**). Color Doppler shows centrally located arterial vessel within the FLL (**b**). Contrast-enhanced ultrasound in the early arterial phase reveals peripheral nodular contrast enhancement (**c**) with almost complete centripetal fill (**d**). In the portal venous phase homogeneously hyperenhancement (**e**). No washout in the later phases (**f**) (L11-3U transducer).

**Figure 2 cancers-15-02360-f002:**
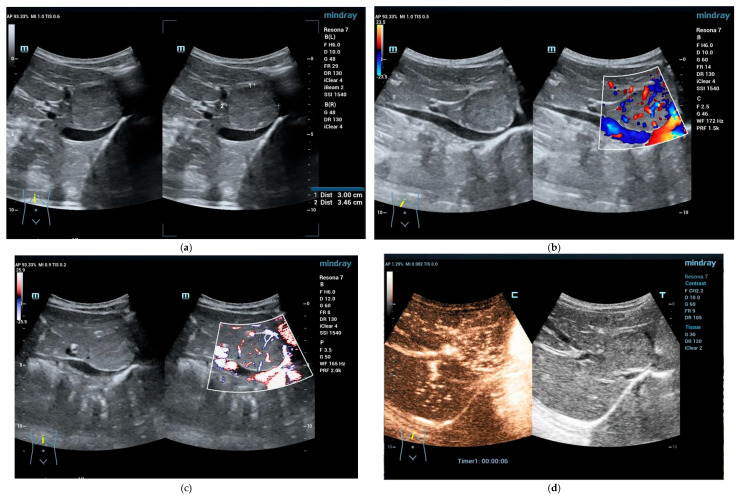
2-year-old girl, incidentally detected FLL. Referred to our hospital for biopsy of the mass. On B-mode 34 mm × 30 mm a well circumscribed homogenous FLL visible in the left lobe (**a**). Color flow Doppler showed centripetal spoke wheel presentation (**b**), more prominent on power Doppler mode (**c**). Contrast-enhanced ultrasound in the early arterial phase reveals very fast arrival time with central artery and spoke wheel pattern (**d**) with complete fill and hyperenhancement a few seconds later (**e**). In the portal venous phase homogeneous isoenhancement (**f**). No washout in the late phase (**g**). Focal nodular hyperplasia diagnosed; biopsy declined. On follow-up, the mass remains the same in three years (**h**) (SC5-1U transducer).

**Figure 3 cancers-15-02360-f003:**
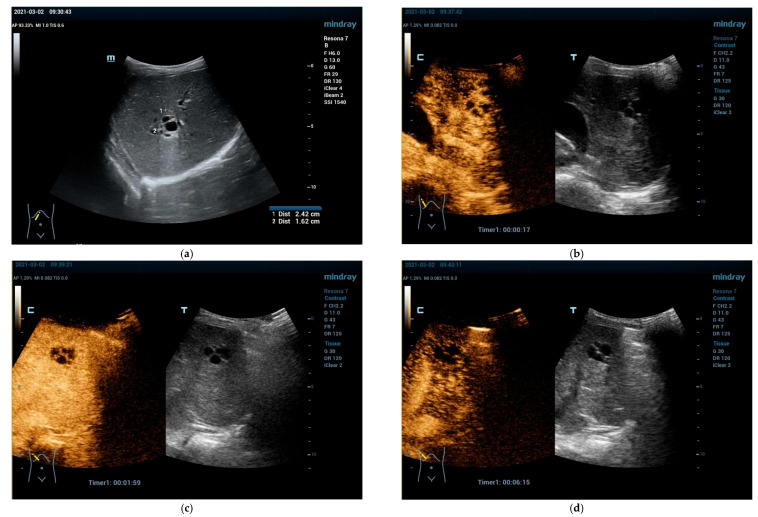
9-year-old girl with incidentally detected liver cyst. B-mode—24 mm × 16 mm septated cyst with echogenic rim in the central part of the liver (**a**). Contrast-enhanced ultrasound in the arterial phase shows only rim and septal enhancement without any flow in the cystic parts (**b**). The same pattern remains in the portal venous (**c**) and late (**d**) phases and no washout in the rim or septa. No treatment nor other investigation suggested (SC5-1U transducer).

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
