# Peer review of "Incidental Findings in Pediatric Patients: How to Manage Liver Incidentaloma in Pediatric Patients"

_cancers, 2023, doi:10.3390/cancers15082360_

Round 1

Reviewer 1 Report

This is an interesting review on an important and emerging problem related to liver incidentalomas which, as stated by actual literature, is commonly encountered with a frequency of up to 33% of radiological studies and 50% of autopsies in adult patients but rarely reported in children.

Clinical scenarios and role of ultrasound and CEUS is very well described and detailed; reported references are updated and complete.

I found extremely interesting and useful sections #3 to #6 which are in line with the Authors statement, underlining that this paper should/could also help the paediatricians and ultrasound practitioners in discriminating pathologies not needing for further investigation, from those requiring interval imaging or immediate additional diagnostic procedures.

Conclusions are appropriate and the proposal of a combined approach including all available clinical information, multiparametric imaging and healthcare and economic impact of every imaging modality is extremely important.

Minor suggestion:

I would suggest the Author to take into consideration this possible inclusion in the following sentence: “Unnecessary further investigation and (over-)treatment may result in a potentially injurious and expensive (this latter depending on specific health care organization insurance systems e.g. Beveridge Model, Bismarck Model, NH Insurance Model, Out-of-pocket payments, etc.) cascade of tests and procedures” (lines 692-693).

Very minor suggestion:

Please, delete dot from sentence “β-. catenin-activated HCA inflammatory” (line 409)

Please, add a space after the dot in the sentence “(GSD 1).HCA are also more commonly associated with diabetes..” (line 398).

Author Response

Dear Editors, dear reviewers,

We appreciate the reviews, which helped us to improve the paper. We hope that the revised paper could be accepted in Cancers.

Cordially

Christoph F Dietrich, corresponding author

This is an interesting review on an important and emerging problem related to liver incidentalomas which, as stated by actual literature, is commonly encountered with a frequency of up to 33% of radiological studies and 50% of autopsies in adult patients but rarely reported in children. Clinical scenarios and role of ultrasound and CEUS is very well described and detailed; reported references are updated and complete. I found extremely interesting and useful sections #3 to #6 which are in line with the Authors statement, underlining that this paper should/could also help the pediatricians and ultrasound practitioners in discriminating pathologies not needing for further investigation, from those requiring interval imaging or immediate additional diagnostic procedures. Conclusions are appropriate and the proposal of a combined approach including all available clinical information, multiparametric imaging and healthcare and economic impact of every imaging modality is extremely important.

Response: Dear reviewer, thank you for the kind and motivating words.

Minor suggestion

I would suggest the Author to take into consideration this possible inclusion in the following sentence: “Unnecessary further investigation and (over-)treatment may result in a potentially injurious and expensive (this latter depending on specific health care organization insurance systems e.g., Beveridge Model, Bismarck Model, NH Insurance Model, Out-of-pocket payments, etc.) cascade of tests and procedures” (lines 692-693).

Response: We inserted: Unnecessary further investigation and (over-)treatment may result in a potentially injurious and expensive cascade of tests and procedures, depending on specific health care organization insurance systems e.g., Beveridge Model, Bismarck Model, NH Insurance Model, Out-of-pocket payments, et cetera

Very minor suggestion

Please, delete dot from sentence “β-. catenin-activated HCA inflammatory” (line 409).

Response: Done (thank you!)

Please, add a space after the dot in the sentence “(GSD 1).HCA are also more commonly associated with diabetes..” (line 398).

Response: Done (thank you!)

Reviewer 2 Report

The paper is undoubtedly relevant in the pediatric framework. However, by reading it seems that most evidence are related to the adult population.

1)In revising the paper the authors should clearly define what papers and evidence are referred to pediatric ages(and refer in detail the pediatric/infant ages).

It seems that most clinical tools have been validated only in  adult pupulation, and it is not clear what may be effectively translated in the pediatric population . In particular the conclussion should be more addressed to pediatric population. 2)An additional issue is that there is no reference to the main laboratory biomarkers to support immaging. I recommend the author to consider this paper and quote it (i.e. for hepatoblastoma, hepatocarcinoma) Ferraro et al. Serum α-fetoprotein in pediatric oncology: not a children's tale. Clin Chem Lab Med 2019 May 27;57(6):783-797.3) Spell out some acronyms the first time you report them.( FLL, WFUMB)

Author Response

Dear Editors, dear reviewers,

We appreciate the reviews, which helped us to improve the paper. We hope that the revised paper could be accepted in Cancers.

Cordially

Christoph F Dietrich, corresponding author

The paper is undoubtedly relevant in the pediatric framework. However, by reading it seems that most evidence are related to the adult population.

1) In revising the paper, the authors should clearly define what papers and evidence are referred to pediatric ages (and refer in detail the pediatric/infant ages). It seems that most clinical tools have been validated only in adult population, and it is not clear what may be effectively translated in the pediatric population. In particular the conclusion should be more addressed to pediatric population.

Response: We cite below a few sentences already shown in the submitted text.

“Liver incidentalomas are commonly encountered, with a reported frequency of up to 33% of radiological studies and 50% of autopsies in adult patients (13, 14) but rarely reported in children. … and in staging and follow-up of previously known focal liver lesions (FLL) including pediatric patients with known cancer or in the setting of surveillance programs for chronic liver disease or other conditions that predispose to malignancy (19, 20, 22) or patients who have undergone interventional procedures (e.g., post chemotherapy, or ablation) (23-27). …”

“Benign lesions are much more common than incidentally detected malignant lesions depending on the age of the pediatric patients.” …

We also updated in particular the conclusion.

2) An additional issue is that there is no reference to the main laboratory biomarkers to support imaging. I recommend the author to consider this paper and quote it (i.e., for hepatoblastoma, hepatocarcinoma) Ferraro et al. Serum α-fetoprotein in pediatric oncology: not a children's tale. Clin Chem Lab Med 2019 May 27;57(6):783-797.3)

Response: We inserted the information and reference (page xx, line xx).

Spell out some acronyms the first time you report them (FLL, WFUMB)

Response: Done

Reviewer 3 Report

The presented work shows both high medical expertise and clinical routine of the authors regarding conventional and contrast-enhanced ultrasound.

Congratulations to the authors for this comprehensive and very understandable review.

Author Response

The presented work shows both high medical expertise and clinical routine of the authors regarding conventional and contrast-enhanced ultrasound. Congratulations to the authors for this comprehensive and very understandable review.

Response: Dear reviewer, thank you for the kind and motivating words.

Reviewer 4 Report

I thank you for the opportunity to review this article. The authors wrote an interesting summary on the role of ultrasound and CEUS in particular for the diagnosis of liver indicentaloma in children. Some english language editing should be performed. Apart from that, I congratulate the authors for this interesting review article. 

Author Response

I thank you for the opportunity to review this article. The authors wrote an interesting summary on the role of ultrasound and CEUS in particular for the diagnosis of liver incidentaloma in children. Some English language editing should be performed. Apart from that, I congratulate the authors for this interesting review article. 

Response: Dear reviewer, thank you for the kind and motivating words. The English was edited by native speakers.

Round 2

Reviewer 2 Report

No further comments